# Genomics-based approach for detection and characterization of SARS-CoV-2 co-infections and diverse viral populations

Bryan Jimenez-Araya,[1] Aurélie Gourgeon,[2] Mélissa N'Debi,[1,3] Taylor Thompson,[1] Vanessa Demontant,[3] Axel Simitambe,[3] Michel Lau,[3] Laure Boizeau,[1,3] Patrice Bruscella,[1] Pierre Cappy,[1,2,3] Jean-Michel Pawlotsky,[1,2,3] Slim Fourati,[1,2,3] Christophe Rodriguez[1,2,3]

**ABSTRACT**  Due to the continuous genetic diversification of severe acute respiratory syndrome coronavirus type 2 (SARS-CoV-2) over time, the co-circulation of two different lineages in the same region may lead to co-infections within a host, a situation known to contribute to the emergence of hybrid viral populations through genomic recombination. The aim of this study was to use a genomics-based approach to identify distinct viral populations of SARS-CoV-2 in patients with coronavirus disease 2019 (COVID-19), as an indicator of potential co-infections and recombination events. The cohort included 41,224 serial nasopharyngeal swabs positive for SARS-CoV-2 RNA, prospectively collected between January 2021 and April 2022 as part of the French national surveillance program. Full-length genomes were sequenced by next-generation sequencing (COVIDseq). Intra-host single nucleotide variants (iSNVs) were identified, and a synthetic cohort was generated to establish thresholds of co-infection detection. Eight hundred sixty-one samples with iSNV ratios above the threshold were considered "potential co-infections." Peaks in co-infection prevalence occurred during the periods of co-circulation of different SARS-CoV-2 variants. Co-infection with different Variants of Concern (VoC) was confirmed in 103 cases, including Alpha-Beta in 12 cases, Alpha-Delta in 15 cases, Gamma-Delta in 4 cases, Delta-Omicron in 35 cases, and Omicron BA.1-BA.2 in 37 cases. In conclusion, our study suggests a higher prevalence of SARS-CoV-2 variant/subvariant co-infection events than that previously reported using conventional approaches, particularly during periods characterized by the emergence and co-circulation of multiple lineages, creating an increased risk of recombination. Our results support the premise of the importance of genomics-based approaches to detect co-infection events in virus-infected populations, including co-infection with closely related lineages.

**IMPORTANCE**  We aim to implement an innovative approach to monitor and study the diversity of severe acute respiratory syndrome coronavirus type 2 (SARS-CoV-2) within the human population, particularly during periods of emergence and circulation of VOCs. This approach focused on detecting highly diverse viral samples and co-infection cases, which are known to facilitate viral diversity through recombination and can potentially lead to the emergence of new recombinant lineages with novel characteristics. Monitoring and characterizing co-infection cases during an outbreak is a key strategy for better understanding viral evolution, especially during epidemic periods. However, detecting co-infection cases is challenging, and their prevalence is often highly underestimated. In this study, we developed a strategy to identify highly diverse viral samples that can be implemented in surveillance programs and applied to large datasets. We aim to implement an innovative approach to monitor and study the diversity of SARS-CoV-2 within the human population, particularly during periods of emergence and circulation of Variants of Concern. This approach focused on detecting highly diverse viral samples and co-infection cases, which are known to facilitate viral

**Peer Reviewer** Aiping Wu, Chinese Academy of Medical Sciences & Peking Union Medical College, Suzhou, China

Address correspondence to Bryan Jimenez-Araya, bryan-stiven.jimenez-araya@inserm.fr.

J.-M.P. has served as an advisor and/or speaker for Abbott, AbbVie, Gilead, and GSK. C.R. has served as a speaker for Pfizer. S.F. has received grants from Moderna and served as a speaker for GlaxoSmithKline, AstraZeneca, MSD, Pfizer, Cepheid, and Moderna. All other authors declare no competing interests.

diversity through recombination and can potentially lead to the emergence of new recombinant lineages with novel characteristics.

**KEYWORDS** SARS-CoV-2, co-infections, next-generation sequencing, viral genomic surveillance

The zoonotic RNA virus severe acute respiratory syndrome coronavirus type 2 (SARS-CoV-2), the causative agent of coronavirus disease 2019 (COVID-19), has spread massively worldwide, causing millions of infections and deaths since it was first reported in December 2019 (1). Since its emergence, mutations and recombination events have led to genomic diversification into new viral lineages, which have subsequently been selected for novel traits such as enhanced transmissibility or potential for immune evasion (2–4).

In late 2020, highly mutated viral variants, characterized by a high proportion of non-synonymous mutations, particularly but not exclusively in the spike protein, began to emerge worldwide. Five of these variants, including the Alpha, Beta, Gamma, Delta, and Omicron variants, were successively classified by the World Health Organization as Variants of Concern (VOCs) (5). Global data reports on SARS-CoV-2 genomic diversity, shared through the GISAID initiative, show periods of co-circulation of different VOCs in different regions, particularly in France and other European countries, as shown in the CoVariants online tool (covariants.org) (6).

During their periods of co-circulation, several cases of co-infection by different SARS-CoV-2 VOCs, particularly Delta and Omicron and different Omicron sublineages, have been reported (7–9). These co-infection events could contribute to the emergence of hybrid viruses as a result of genomic recombination, a common mechanism known to drive viral evolution (10).

In coronaviruses, this mechanism is thought to occur through a template switch of the RNA-dependent RNA polymerase (RdRp) during replication in a co-infected cell. This involves the dissociation of the RdRp from one genomic strand and its subsequent association with another, resulting in the synthesis of a recombinant strand, which is also known as "copy-choice replication" (11), which increases the likelihood of viral genomic diversification occurring in co-infected host cells, particularly during an outbreak when multiple viral variants are circulating extensively (12).

Several recombination events of SARS-CoV-2 have been reported, including the description of inter-VOC and intra-Omicron recombinants (13). Most of these recombinants had limited circulation in the population. However, recombinant lineages with improved transmissibility and potential for immune evasion emerged in late 2022 (XBB recombinant and its progeny) and spread globally in 2023 (3, 14). Since then, the trajectory of the SARS-CoV-2 spread has been influenced by several combinations of recombinant lineages. By the end of December 2024, more than 130 Pango-designated recombinant lineages of SARS-CoV-2 with the prefix "X," had been reported (15), demonstrating the continuing role of recombination in SARS-CoV-2 evolution.

The search for SARS-CoV-2 co-infections and recombination events has been limited. The global pandemic context presented many technical and analytical challenges for the effective genomic surveillance and monitoring of SARS-CoV-2 variants (16). Thus, it is plausible that the co-infection rates were severely underestimated during the co-circulation periods, making it crucial to implement effective global genomic surveillance strategies and collaborative efforts in this direction (17, 18).

Here, we used a genomics-based approach to identify distinct viral populations of SARS-CoV-2 in patients with COVID-19, as an indicator of potential co-infection and recombination events.

## RESULTS

### Study population

Between January 2021 and April 2022, 63,195 nasopharyngeal swabs positive for SARS-CoV-2 RNA were sent to our laboratory for next-generation sequencing of the full-length viral genomes as part of the French national surveillance program. Of these, 41,224 samples passed quality controls for full-length genome reconstruction and were included in our study of the diversity of SARS-CoV-2 populations, as shown in the flowchart in Fig. 1. The reconstruction of the full-length viral genome sequences had a median length coverage of 99.2% of the full-length sequence (range: 92% to 100%) and 99.7% of the spike-coding region (range: 92% to 100%). The median sequencing depth was 39.5 full-length sequences (range: 12.2 to 64.6).

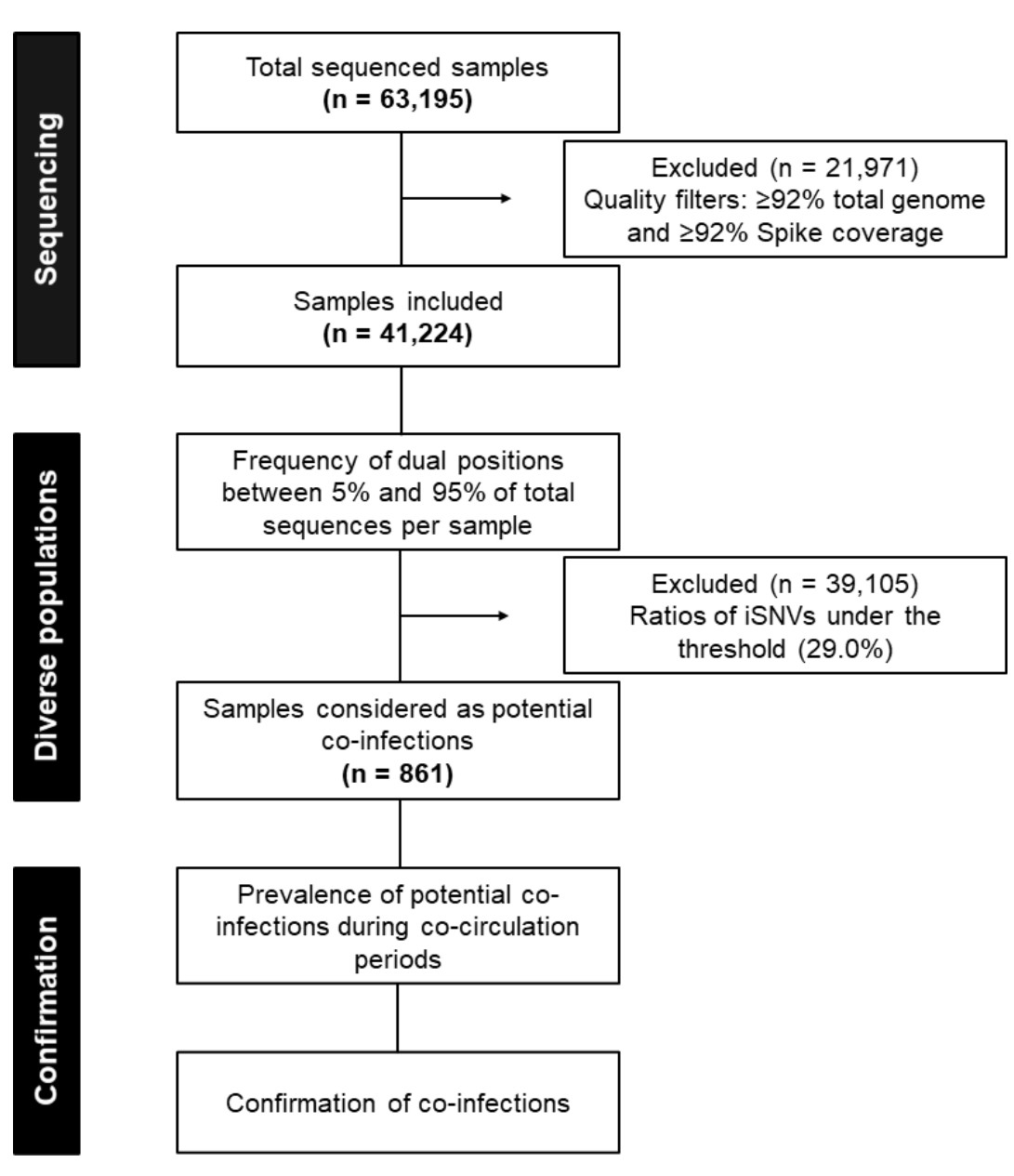

**FIG 1** Flowchart of the study.

## SARS-CoV-2 variant circulation

Classification of the viral sequences into lineages and clades was performed using Pangolin and NextClade. The date of collection was used to describe the circulation dynamics of the different variants during the study period (Fig. 2). At each weekly time point, a mean of 606 ± 403 sequences were available.

During the first 4 weeks of the study (January 2021), the Alpha variant was found in 17% to 41% of cases, circulating simultaneously with older lineages, including Wuhan and others. Between weeks 9 and 23 (March to June 2021), the proportion of the Alpha variant increased to 70%–90%, while the Beta variant represented 10%–30% of cases. The Gamma variant was found in 156 samples between weeks 5 and 36 (February to August 2021). As shown in Fig. 2, the Alpha, Beta, and Gamma variants circulated simultaneously in France for several months.

The Delta variant was first detected in week 17 (May 2021) and its proportion increased rapidly in the following weeks. By week 25 (late June 2021), the Delta variant peaked at 50% of total infections, surpassing the other circulating variants. Between weeks 29 and 49 (late July to early December 2021), the Delta variant accounted for 95% to 99% of cases, while the Alpha, Beta, and Gamma variants gradually disappeared. Our results show that the Delta variant circulated simultaneously with the Alpha, Beta, and Gamma variants for several weeks.

The Omicron variant was first detected in the cohort at week 45 (mid-November 2021) (<1%) and its prevalence increased dramatically by week 51 (end-December 2021). By week 57 (end of January 2022), Omicron represented 99% of cases. Thus, Delta and Omicron variants shared a period of co-circulation between weeks 45 and 60 (November 2021 to February 2022). The proportion of Omicron cases remained at 99% until the end of the study period in April 2022, characterized by the co-circulation of sublineages BA.1 and BA.2, with the latter gradually replacing the former.

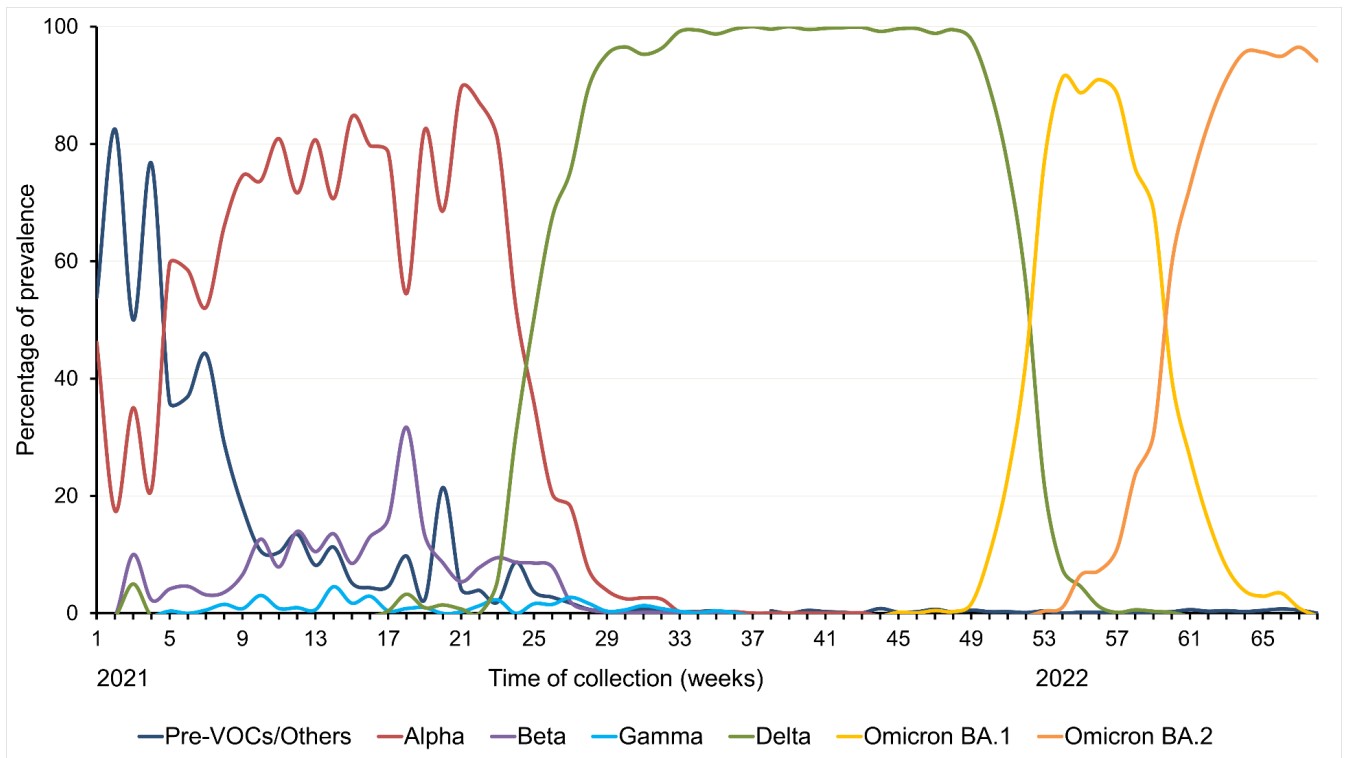

**FIG 2** Dynamics of circulation of SARS-CoV-2 variants during the study period.

## Validation of the threshold for detection of diverse SARS-CoV-2 populations

The presence of diverse SARS-CoV-2 populations in the samples was assessed as described in the Materials and Methods. Intra-host single nucleotide variants (iSNVs), i.e., positions in the genome with more than one nucleotide in the corresponding swab, were identified, and normalized iSNV ratios were calculated.

A cohort of synthetic SARS-CoV-2 genome sequences was generated to validate the method and establish sensitive thresholds for co-infection detection. The cohort consisted of synthetic sequences simulating co-infections by mixing the sequences of different reference VOCs (Alpha-Beta, Alpha-Delta, Delta-Omicron BA.1, and Omicron BA.1-BA.2) in different proportions (5% to 95% genome mixing) and synthetic sequences of pure VOC genomes. The synthetic files were analyzed to determine the iSNV ratios. The analysis showed that highly co-infected samples (mixing ratio of 20% or higher) had iSNV ratios higher than 22.3% ($P < 0.01$), while pure synthetic samples (one variant) and low co-infected samples (mixing ratio below 15%) had iSNV ratios lower than 29.0% ($P < 0.01$) (Fig. 3A). The range between 22.3% and 29.0% was then considered as a gray zone of uncertain iSNV ratios, while a conservative ratio of 29.0% was chosen as the minimum threshold to suggest the presence of co-infection. Using this threshold, a receiver operating characteristics (ROC) curve was constructed that showed an area under the curve (AUC) >99% for synthetic co-infections with a genome mixing ratio ≥20% (Fig. 3B), validating the sensitivity of the method to discriminate highly co-infected samples from negative/low co-infected samples.

## Detection of diverse SARS-CoV-2 populations

Based on these results, a total of 861 samples from our cohort, corresponding to the samples with iSNV ratios greater than 29.0%, were considered potential carriers of co-infections (Fig. 4). Figure 5 shows the prevalence of these 861 nasopharyngeal swabs in relation to the number of samples tested during the study period, according to the co-circulation of different VOCs. Several peaks in the prevalence of potential co-infections were observed over time.

The prevalence of potential co-infections ranged from 0.3% to 2.4% of cases between weeks 4 and 16 (January to April 2021), when the Alpha, Beta, and Gamma variants co-circulated with previous viral lineages. This prevalence ranged from 0% to 4.2% between weeks 17 and 32 (May to August 2021), when the Alpha and Delta variants had a high incidence in the population and the Beta and Gamma variants also co-circulated. Prevalence of potential co-infections ranging from 0.1% to 4.3% was observed between weeks 33 and 48 (August to December 2021), with the Delta variant circulating predominantly (98%–99%). The highest peaks of potential co-infections were observed between weeks 49 and 60 (December 2021 to February 2022), with prevalences ranging from 1.6% to 10.5%, when the Delta and Omicron variants, and then the different Omicron sublineages, co-circulated.

## Confirmation of co-infections

The 861 samples classified as potential co-infection carriers were further analyzed to identify VOC-specific mutations and to confirm the possible presence of more than one variant. Of the 861 potential co-infection carriers, 103 (11.96%) were confirmed to be co-infected with different variants: Alpha and Beta in 12 cases, Alpha and Delta in 15 cases, Gamma and Delta in 4 cases, Delta and Omicron in 35 cases, and Omicron BA.1 and BA.2 in 37 cases. A list of the VOC mutations and iSNV ratios identified in all 103 co-infections is shown in Table S1. Despite their co-circulation in the population, no Alpha-Gamma, Beta-Gamma, or Beta-Delta co-infections were detected in our samples. Furthermore, of the 35 Delta-Omicron co-infections identified, only one corresponded to a Delta-Omicron BA.2 co-infection. A summary of the confirmed co-infections is shown in Table 1.

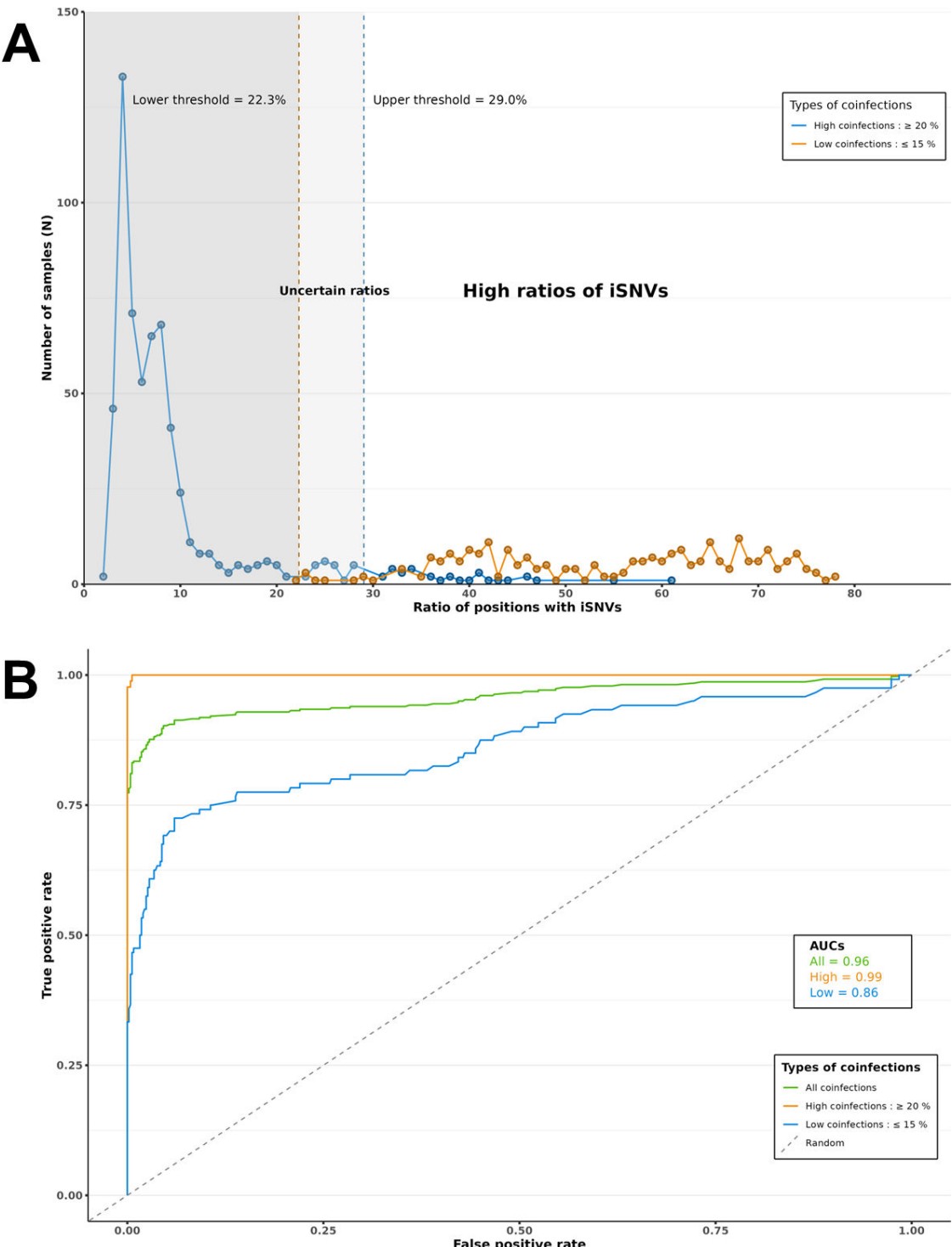

**FIG 3** (**A**) Curve representing the iSNV ratio values of synthetic samples of pure variants and low co-infections (blue curve) and synthetic samples of high co-infections (orange curve). Thresholds are calculated to include true negatives (pure variants) and low co-infections (*P* < 0.01) (blue dotted lines) and true positives greater than or equal to 20% (*P* < 0.01) (orange dotted lines). The ratio of 29.0% was used for the patient data in order to have the minimum number of false co-infections. (**B**) At the threshold of 29.0%, the AUC of the ROC curve shows a value of more than 99% for the classification of a sample as co-infected when the mixture of variants within a sample exceeds 15% (orange curve) and drops only to 86% for lower ratios (blue curve).

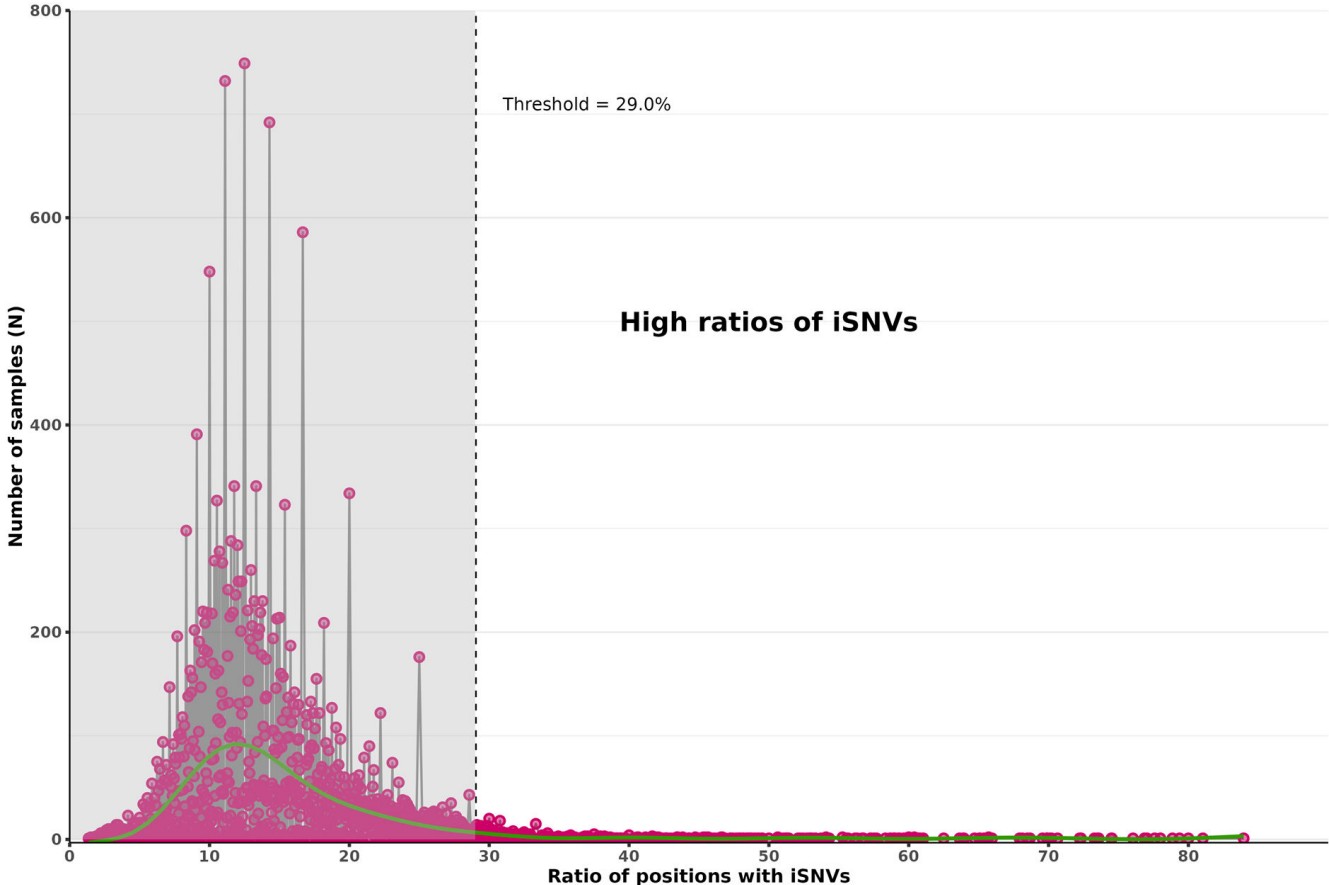

**FIG 4** Distribution of the 41,224 samples according to their ratio of double nucleotides at different positions in the full-length genome. The Gaussian distribution was assumed to be due to sequencing errors. Thus, only the samples ($n = 861$) with ratios higher than 29.0% (threshold obtained from the synthetic data test) were considered as potential carriers of co-infections. The smooth curve is shown in green as a visual aid with reduced noise. The pink points represent the number of samples (y-axis) with a given ratio (x-axis).

## DISCUSSION

Our data on the circulation of the VOCs Alpha, Beta, Gamma, Delta, and Omicron closely match those reported at the national level in France (6), reflecting the very large size of our cohort and the diversity of origin of the samples tested by our platform during the pandemic as part of the national EMERGEN surveillance program. We therefore consider our data to be representative of the French national situation. As shown here and confirmed by official reports, several periods of simultaneous circulation of two or more VOCs followed each other in the French population. These periods are known to be crucial for the genomic diversification of the virus and the possible emergence of recombinant variants due to co-infection events (5).

Previous studies have reported co-infection rates during different periods of the COVID-19 pandemic, including with pre-Alpha lineages and subsequent VOCs ranging from 0.18% to 0.6% in several studies (7, 9, 18–21), and a striking 5% in one study during the pre-VOC era (22). In the present study, we report that 2% of samples (861 out of 41,224) in a French national cohort are likely to carry SARS-CoV-2 co-infections. In addition, a general rate of co-infection between VOCs of 0.25% (103 co-infections in 41,224 samples) was confirmed, which is consistent with the range of rates reported from other countries.

In particular, the Delta-Omicron and Omicron BA.1-BA.2 combinations had the highest rates in our cohort, at 0.27% and 0.36%, respectively. Similarly, two other French studies have reported co-infection rates of 0.20% and 0.18% for Delta-Omicron, and one

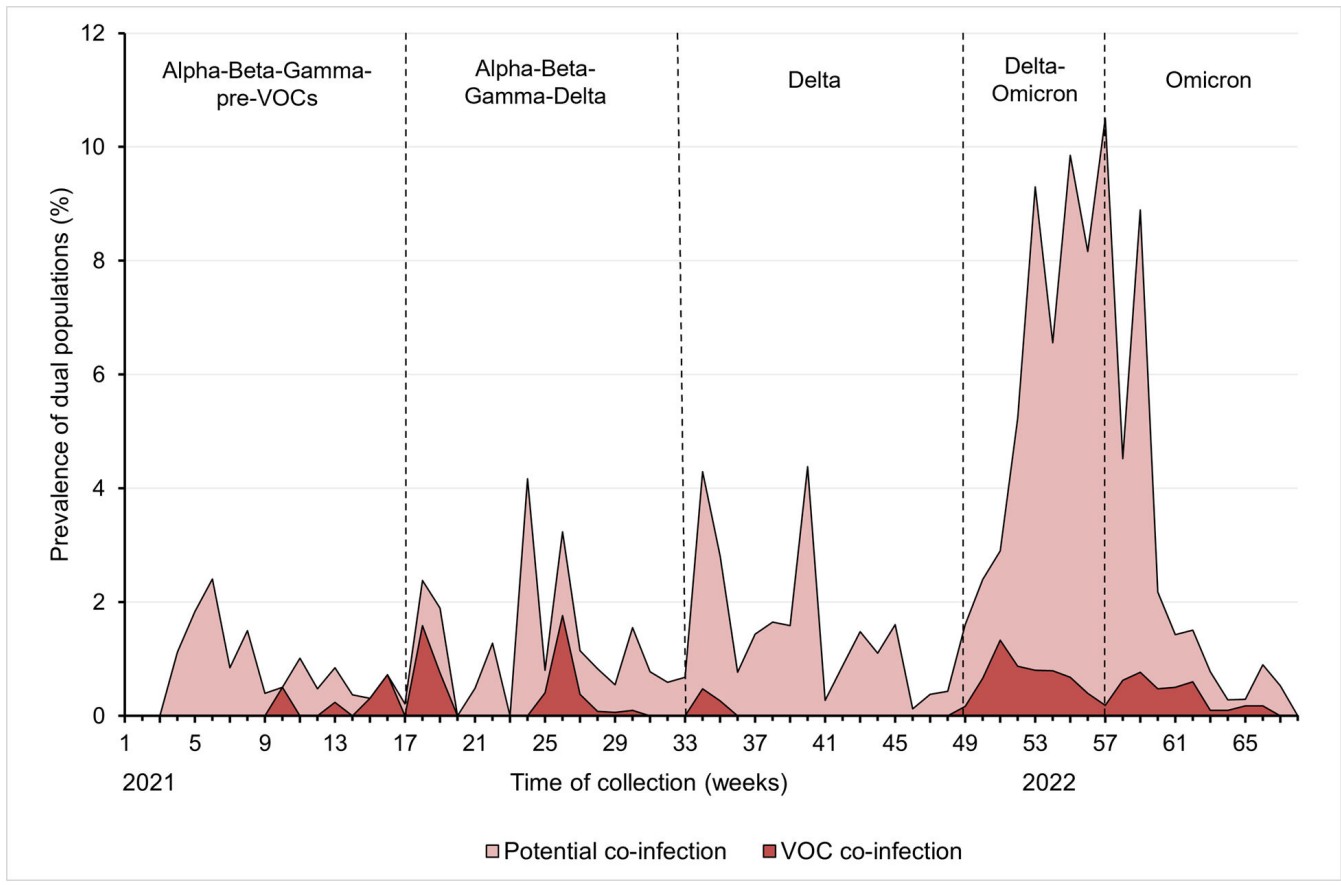

**FIG 5** Trend in the prevalence of nasopharyngeal swab samples with potential and confirmed co-infection relative to the number of samples tested during the study period, according to the co-circulation of different VOCs.

of the studies reported a rate of 0.26% for Omicron BA.1-BA.2 (7, 9). Although these rates are slightly lower than those observed in our study, they indicate comparable national trends of co-infection with these variants.

As SARS-CoV-2 is known to have low levels of intra-host diversity (23), we used an approach based on the detection of high iSNV ratios to separate the samples based on their genomic diversity, with the aim of detecting true co-infections and characterizing VOC combinations. We used synthetic data, including *in silico* generated co-infections, to determine thresholds for true co-infection detection and to minimize identification errors. This approach allowed us to identify highly divergent samples from our cohort, including samples containing SARS-CoV-2 co-infections. However, the identification of low levels of co-infection (below 15% mixing) was found to be inaccurate with this method. Therefore, a threshold for accurate detection of highly co-infected samples was established to avoid false positives. As the minimum threshold for detection of

**TABLE 1** Summary of the 103 confirmed co-infections identified in the cohort

| VOCs | Number of confirmed co-infections | Number of cases during the co-circulation period | Co-infection rate (%) |
|---|---|---|---|
| Alpha-Beta | 12 | 6,505 | 0.18 |
| Alpha-Delta | 15 | 11,489 | 0.13 |
| Gamma-Delta | 4 | 9,530 | 0.04 |
| Delta-Omicron | 35 | 13,145 | 0.27 |
| Omicron BA.1-BA.2 | 37 | 10,339 | 0.36 |
| General rate | 103 | 41,224 | 0.25 |

highly co-infected samples was 22.3%, meaning that the 29.0% threshold used may have excluded a fraction of the true co-infected samples, our cohort may have slightly underestimated the reported co-infection rates.

A potential bias of this method is the significantly higher intra-host genetic diversity of the virus in immunocompromised individuals (24, 25). As these patients often have persistent infections, long-term viral replication can lead to the accumulation of mutations and the emergence of highly mutated variants, influencing viral evolution (26, 27). However, immunocompromised patients are very rare in large national programs such as the one used here and are therefore unlikely to influence the overall results.

Our results demonstrate the existence of co-infections during the periods of co-circulation of different VOCs and Omicron sublineages, a favorable condition for the evolution of novel recombinant variants. Genomic surveillance programs have not identified such co-infections and have assigned variants from the consensus genome sequence. Thus, the use of consensus genomic data may result in a loss of important information by preventing the identification of minor variants present in samples, leading to an underestimation of co-infections during periods of co-circulation of different viral variants.

Our study included a very large number of samples and used an approach that specifically targeted the differentiating mutations of each variant. This approach allowed us to generate key information for a better understanding of the variability of the virus and to observe the changing rates of VOC co-infection during different periods of co-circulation of these variants. However, this approach limited our analysis of co-infections to the major, dominant lineages, potentially underestimating co-infections with more closely related SARS-CoV-2 lineages or sublineages, that may also play a role in the genetic evolution of the virus.

For example, our approach did not target mutations from pre-VOC lineages circulating in early 2021. Several peaks of high iSNV ratios were identified during this period that did not meet our previously established criteria for confirmation of co-infections, which was limited to Alpha-Beta and later to Alpha-Delta and Gamma-Delta co-infections. No co-infections were detected during the predominant circulation of Delta VOC, as no other target variants were circulating. Nevertheless, we detected peaks of high iSNV ratios that could not be attributed to confirmed co-infection events. We hypothesize that a significant number of co-infections with different Delta sublineages occurred, a hypothesis supported by previous reports of Delta-Delta co-infections (21). Finally, our approach only targeted Omicron BA.1 and BA.2 mutations, limiting our ability to detect other combinations. This may explain the discrepancy between the high iSNV ratio peaks and the relatively low number of confirmed co-infections.

The increase in iSNV ratios during the period of Omicron variant emergence is consistent with the occurrence of multiple lineage and sublineage co-infections. This hypothesis is supported by the observation of the emergence of an increasing number of recombinant variants around the study period, including reports of 5 Delta-Omicron (17, 28) and over 30 intra-Omicron recombinant lineages (17). These findings highlight the importance of continuously developing strategies for monitoring co-infection and genetic recombination, both for currently circulating viruses and for potential future emergences.

Since then, several intra-Omicron recombinant lineages have continued to emerge, including lineages such as XBB that have spread rapidly around the world (29, 30), resulting in a total of approximately 130 SARS-CoV-2 recombinant lineages reported to date (15). In addition, certain recombinant lineages have shown increased transmissibility, viral fitness, and immune evasion. This has led to significant concerns about the reduced effectiveness of existing vaccines and the widespread global spread of these lineages (30, 31).

Our study has several limitations. (i) Although the use of synthetic data gave promising results in terms of defining thresholds for considering a sample to be potentially co-infected, these are still artificially generated files that could lead to

inaccuracies. Therefore, the term "potential co-infection" was chosen. (ii) Our sequencing method did not provide single-cell resolution. Since recombination events require co-infection of the same cell, the presence of co-infection in our patients was taken as an indicator of plausible viral recombination. (iii) Low sequencing depth and sequencing errors could lead to misdetection of iSNVs. However, the detection criteria used in this study (at least three copies of the minor allele(s) and a frequency greater than 5%) probably prevented misidentification in most cases. (iv) The use of a few selected VOC-specific mutations to confirm co-infections could have led to misidentifications. However, the presence of at least two mutations specific for a second variant gave a clear indication of the genomic diversity of the sample, even if the detected sequence was from a recombinant variant rather than a co-infection, thus excluding the possibility of false positives due to sequencing errors. (v) Confirmation of the presence of minor lineages was not straightforward due to the large number of possible closely related lineages and sublineages co-circulating and may require a more detailed approach for this purpose. However, the identification of every single minor lineage was not part of the objectives of this study.

In conclusion, the method using iSNV data described here is a promising approach for the detection of highly diverse viral samples and co-infection. Our study suggests a higher prevalence of SARS-CoV-2 variant/subvariant co-infection events than that previously reported using conventional approaches, particularly during periods characterized by the emergence and co-circulation of multiple lineages, leading to an increased risk of recombination. The emergence of new viral variants is a major public health concern, affecting viral transmission rates, virulence, evasion of host immunity, and the efficacy of previous antiviral strategies such as vaccines and therapeutics. It is therefore crucial to strengthen surveillance systems by monitoring co-infection cases and the overall dynamics of viral evolution in the context of an epidemic. Our results support the premise of the importance of genomics-based approaches to detect co-infection events in virus-infected populations, including co-infections with closely related lineages. They support the future use of the approach used in our study to predict and prevent the emergence of recombinant viral variants.

## MATERIALS AND METHODS

### Study design

This is an epidemiological study based on viral genomic data collected prospectively during the COVID-19 pandemic in France. The material studied here consisted of 63,195 nasopharyngeal swabs from COVID-19 patients sent to the Genomic Platform of the Henri Mondor University Hospital for viral genome sequencing and analysis. The samples were collected between January 2021 and April 2022 in different regions of France as part of the French EMERGEN COVID-19 pandemic surveillance program. All samples included were confirmed for SARS-CoV-2 infection by a positive reverse transcriptase-polymerase chain reaction result.

### Viral genome sequence analysis

The full-length SARS-CoV-2 genomes of all enrolled subjects were sequenced by means of next-generation sequencing. Briefly, viral RNA was extracted from nasopharyngeal swabs in viral transport medium using Chemagic Viral DNA/RNA 300 Kit H96 on a Chemagic Prime instrument (Perkin Elmer, Waltham, Massachusetts, USA). Sequencing was performed using the Illumina COVIDSeq Test (Illumina, San Diego, California, USA), which uses a 98-target multiplex amplification along the entire SARS-CoV-2 genome. Libraries were sequenced using the NovaSeq 6000 SP Reagent Kit v1.5 (100 cycles) on a NovaSeq6000 instrument (Illumina). Sequences were demultiplexed and assembled as full-length genomes using the DRAGEN COVIDSeq Test Pipeline on a local DRAGEN server (Illumina). Phylogenies and clades were interpreted for all samples using Pangolin (v3.1.17) and NextClade (v1.4.2).

Only the 41,224 samples with ≥92% coverage of the full-length SARS-CoV-2 genome sequence and ≥92% coverage of the spike protein coding region were included in the final analysis. All sequences were submitted to the GenBank database (accession numbers PQ968048–PQ997909).

## Detection of diverse SARS-CoV-2 populations

To identify the SARS-CoV-2 populations present in the samples, full-length genome sequences were aligned to the SARS-CoV-2 reference genome sequences using an in-house pipeline, and variant calling was performed using the Ivar software (v1.3.1).

iSNVs, i.e., positions in the genome with more than one nucleotide in the corresponding swab, were identified. Only samples with at least 10 reads and iSNVs with at least three copies of the minor allele(s) and a frequency greater than 5% were included. To obtain the iSNV ratios, the number of iSNVs present in each sample was divided by the total number of mutations in the full-length sequence of the corresponding assigned variant for normalization. The positivity ratio for reporting potential co-infection is based on the synthetic data test, described in the next section.

## Validation of co-infection thresholds from synthetic data

Synthetic sequence files were generated from SARS-CoV-2 reference genomes obtained from GenBank for the Alpha (ID: MZ344997.1), Beta (ID: MW598419.1), Delta (ID: MZ009823.1), Omicron BA.1 (ID: OL672836.1), and Omicron BA.2 (ID: OM371884.1) variants, including synthetic co-infection files using different VOC combinations and proportions. These files were generated (i) by fragmentation of the reference genome (280 ± 30 bp) to simulate the size of the experimental workflow; (ii) by generating reads (36 bp) from the synthetic fragments; (iii) by introducing error by randomly selecting a Q-score among 12, 23, and 37 (with distributions of 0.034, 0.026, and 0.940, respectively); and (iv) by generating FASTQ files by mixing the synthetic reads from each variant in the specified proportions (example: 5% reads from Alpha and 95% reads from Beta). The combinations chosen were Alpha-Beta, Alpha-Delta, Delta-Omicron BA.1, Omicron BA.1-BA.2. Five replicates of each mixing percentage were performed for each variant combination, resulting in 380 files. In addition, 100 files of pure variant sequences were generated, giving a total of 880 files.

The files contained a total of 32,400 ± 1,000 reads. They were analyzed under the same conditions as the patient samples used to calculate the iSNV ratio. Two types of synthetic files were analyzed: (i) "high co-infections" (20% to 80% genomic mixtures), and (ii) pure variants (0%) and low co-infections (5% to 15% genomic mixtures). Thresholds for the detection of potential co-infections were obtained using the formula mean of the ratios + 2.326 × standard deviation of the ratios, where 2.326 is the Student's coefficient for a one-tailed test with 99% confidence ($P < 0.001$). The lower threshold corresponds to the lowest iSNV ratio for detecting true positives from the files containing ≥20% variant mixtures. The higher threshold corresponds to the highest iSNV ratio for the detection of true negatives (pure variants) and co-infections with <15% genome mixtures. Once the threshold was defined, a ROC curve was plotted and the AUC was calculated to determine the sensitivity of the algorithm for detecting co-infections. Both the ROC curve and the AUC were calculated using an R script and the "ROCR" package. Briefly, each synthetic sample was tagged (highly co-infected vs pure/low co-infection) and the ratios generated by the analysis were associated with these tags. A prediction was made, and performance was calculated using true positive and false positive rates.

## Confirmation of co-infections

The samples with iSNV ratios above the threshold were considered potential containers of co-infections and further analyzed to identify the presence of specific VOC co-infections. For this purpose, a list of VOC-specific mutations was selected from the lineage comparison tool of the Outbreak.info genomic reports (32), specifically in the spike

region of the genome, as this region is known to be highly variable among variants and to contain hotspots of mutations that allow differentiation of VOCs (33, 34). These mutations were selected on the basis of their reported specificity for the corresponding VOC and their absence in other co-circulating VOCs. These mutations were used to determine whether the samples had true co-infections. The spike mutations of interest were (i) Alpha VOC: A570D (C23271A), T716I (C23709T), S982A (T24506G), D1118H (G24914C); (ii) Beta VOC: D80A (A21801C), D215G (A22206G), A701V (C23664T); (iii) Gamma VOC: L18F (C21614T), T20N (C21621A), D138Y (G21974T), R190S (G22132T), K417T (A22812C), T1027I (C24642T), V1176F (G25088T); (iv) Delta VOC: T19R (C21618G), L452R (T22917G), P681R (C23604G), D950N (G24410A); (v) Omicron BA.1 VOC: N211I (A22194T), T547K (C23202A), N856K (C24130A), L981F (C24503T); (vi) Omicron BA.2 VOC: T19I (C21618T), L24S (T21633C), V213G (T22200G), S371F (C22674T). To be considered co-infected, samples must have at least two of the selected VOC-specific mutations of another variant. Co-infection rates were calculated for each VOC combination detected by dividing the number of samples confirmed as co-infected by the total number of cases during the period of co-circulation of the two variants.

GPT-4

## ACKNOWLEDGMENTS

The authors thank all the EMERGEN partners, the diagnostic laboratories that voluntarily selected and sent their samples for sequencing, and the patients. The artificial intelligence tools ChatGPT (OpenAI's GPT4GPT-4, https://chatgpt.com/) and DeepL Write (DeepL, https://www.deepl.com/fr/write) were used to improve the English language in the final manuscript.

This work was funded by the French government (Ministry of Health and Caisse Nationale d'Assurance Maladie) as part of the EMERGEN national COVID-19 molecular surveillance program.

B.J.-A. coordinated the study, supervised the data analysis, and data interpretation, and drafted the first version of the manuscript. A.G., M.N., and T.T. performed the statistical analysis and managed and interpreted the data. V.D., A.S., and M.L. performed the COVIDseq technique. L.B. contributed to data analysis and data interpretation. P.B., P.C., and J.-M.P. revised and edited the manuscript. S.F. and C.R. coordinated the study, interpreted the COVIDseq results, and revised and edited the manuscript. All authors contributed substantially to the manuscript and approved the final version.

## AUTHOR AFFILIATIONS

[1]Team "Viruses, Hepatology, Cancer", INSERM U955 (IMRB), University of Paris-Est, Créteil, France
[2]Department of Virology, Henri Mondor University Hospital (AP-HP), University of Paris-Est, Créteil, France
[3]GenoBIOMICS platform, Henri Mondor University Hospital (AP-HP), University of Paris-Est, Créteil, France

## AUTHOR ORCIDs

Bryan Jimenez-Araya  http://orcid.org/0009-0006-1962-0666
Jean-Michel Pawlotsky  http://orcid.org/0000-0003-0745-7559
Slim Fourati  https://orcid.org/0000-0002-1236-5467

## AUTHOR CONTRIBUTIONS

Bryan Jimenez-Araya, Conceptualization, Project administration, Supervision, Writing – original draft | Aurélie Gourgeon, Conceptualization, Data curation, Formal analysis, Methodology, Software, Writing – review and editing | Mélissa N'Debi, Conceptualization, Data curation, Formal analysis, Methodology, Software, Writing – review and editing | Taylor Thompson, Formal analysis, Methodology, Writing – review and editing | Vanessa

Demontant, Methodology, Writing – review and editing | Axel Simitambe, Methodology, Writing – review and editing | Michel Lau, Methodology, Writing – review and editing | Laure Boizeau, Formal analysis, Methodology, Writing – review and editing | Patrice Bruscella, Conceptualization, Writing – review and editing | Pierre Cappy, Supervision, Writing – review and editing | Jean-Michel Pawlotsky, Funding acquisition, Supervision, Writing – review and editing | Slim Fourati, Project administration, Supervision, Writing – review and editing | Christophe Rodriguez, Funding acquisition, Project administration, Supervision, Writing – review and editing

## DATA AVAILABILITY

All sequences were submitted to the GISAID database as part of the national surveillance program. GenBank accession numbers specifically for the samples used in the cohort are PQ968048–PQ997909, containing the consensus genomes. In addition, the raw sequence data (FASTQ files) from the next generation sequencing are available to be shared by demand (approximate size of the total dataset is 2To).

## ETHICS APPROVAL

The authors declare that this report contains no personal information that could lead to identify patients and/or volunteers. Respondents were informed of the purpose of the study and their right to refuse, and interviewers anonymized the data before passing them on to the team responsible for the analysis. All analyses were performed on pseudonymized or anonymized data, in accordance with the legal and regulatory prerogatives granted to the EMERGEN French COVID-19 molecular surveillance program to fulfill its public interest mission, and in compliance with the provisions of the General Data Protection Regulation (GDPR). In this context, the opinion of an ethics committee was not required.

## ADDITIONAL FILES

The following material is available online.

### Supplemental Material

**Table S1 (Spectrum02092-24-s0001.docx).** VOC-specific mutations and iSNV ratios present in confirmed SARS-CoV-2 co-infections.

### Open Peer Review

**PEER REVIEW HISTORY (review-history.pdf).** An accounting of the reviewer comments and feedback.

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
