## [Reviewer comments · Microbiology Spectrum]

Microbiology Spectrum

Genomics-based approach for detection and characterization of SARS-CoV-2 co-infections and diverse viral populations

Bryan Jimenez-Araya, Aurélie Gurgeon, Mélissa N'Debi, Taylor Thompson, Vanessa Demontant, Axel Simitambe, Michel Lau, Laure Boizeau, Patrice Bruscella, Pierre Cappy, Jean-Michel Pawlotsky, Slim Fourati, and Christophe Rodriguez

Corresponding Author(s): Bryan Jimenez-Araya, Institut Mondor de recherche biomédicale

Review Timeline:

Submission Date:	August 20, 2024
Editorial Decision:	October 24, 2024
Revision Received:	January 27, 2025
Accepted:	February 13, 2025

Editor: Day-Yu Chao

Reviewer(s): Disclosure of reviewer identity is with reference to reviewer comments included in decision letter(s). The following individuals involved in review of your submission have agreed to reveal their identity: Aiping Wu (Reviewer #2)

Transaction Report:

DOI: <https://doi.org/10.1128/spectrum.02092-24>

Re: Spectrum02092-24 (Dual SARS-CoV-2 infections: Genomics-based approach for detection and characterization of SARS-CoV-2 co-infections and diverse viral populations)

Dear Dr. Bryan Jimenez-Araya:

Thank you for the privilege of reviewing your work. Below you will find my comments, instructions from the Spectrum editorial office, and the reviewer comments.

Revision Guidelines

Sincerely,
Day-Yu Chao
Editor
Microbiology Spectrum

Reviewer #1 (Comments for the Author):

Please see attached document.

Reviewer #2 (Comments for the Author):

The manuscript presents a genomics-based study investigating the prevalence and characteristics of dual SARS-CoV-2 infections (co-infections) in France between January 2021 and April 2022. Analyzing 41,224 nasopharyngeal swab samples, the authors employed next-generation sequencing to detect intra-host single nucleotide variants (iSNVs) as indicators of potential co-infections. They identified 861 samples with significant iSNV ratios and confirmed 104 cases of co-infections involving different variants of concern (VOCs), including combinations like Alpha-Beta, Alpha-Delta, Gamma-Delta, Delta-Omicron, and Omicron BA.1-BA.2. The study suggests that co-infections were more prevalent than previously reported, especially during periods of co-circulation of multiple VOCs, highlighting the importance of genomics-based surveillance to detect and monitor such events for better understanding viral evolution and mitigating the risk of recombinant variants.

Major Comments:

1. Clarity and Detailed Description of Methodology:

iSNV Detection and Threshold Determination: The methods section would benefit from a more comprehensive explanation of how iSNVs were identified and the rationale behind choosing the specific thresholds (e.g., frequency greater than 5%, $p < 0.01$ for iSNV ratios). Providing details on how the background noise was calculated and how the threshold of 29.05% was established would enhance reproducibility and understanding.

Confirmation of Co-infections: The criteria for confirming co-infections based on VOC-specific mutations in the spike protein should be elaborated. Discussing the potential for false positives or negatives when only a subset of mutations is used for confirmation would strengthen the validity of the findings.

2. Statistical Analysis and Interpretation:

Significance Testing: More detailed information on the statistical methods used to assess the significance of observed iSNV ratios would be valuable. For instance, explaining the statistical tests applied and any assumptions made would improve the robustness of the results.

Co-infection Rates Calculation: Clarify how the co-infection rates were calculated concerning the total number of cases during co-circulation periods. Including confidence intervals would provide a sense of the precision of these estimates.

3. Discussion of Limitations:

The manuscript would benefit from a more in-depth discussion of the limitations inherent in the study. For example, acknowledging potential biases due to sample selection, limitations in detecting co-infections with closely related lineages, and the impact of sequencing depth and quality on iSNV detection would provide a balanced perspective.

Consider discussing the potential influence of immunocompromised patients on iSNV rates, as they may exhibit higher intra-host diversity, potentially confounding the detection of co-infections.

4. Data Accessibility and Transparency:

While the authors mention that sequences were submitted to the GISAID database and that raw data are available upon request, providing accession numbers or a direct link to the dataset would enhance transparency and allow other researchers to verify and build upon the work.

Including supplemental materials with additional data tables, such as a detailed list of confirmed co-infection cases and their associated mutations, would be beneficial.

5. Contextualization with Existing Literature:

The discussion could be enriched by comparing the co-infection rates found in this study with those reported in other regions or countries, exploring reasons for similarities or differences.

Incorporating recent literature on recombinant variants and their impact on public health would emphasize the relevance of the findings.

Minor Comments:

1. Consistency in Terminology:

Ensure consistent use of terms such as "co-infection," "dual infection," "mixed infection," etc., throughout the manuscript to avoid confusion.

2. Enhance the Introduction:

Provide a more detailed background on the mechanisms of SARS-CoV-2 recombination and why co-infections are pivotal in this context. This would set a stronger foundation for the study's significance.

3. Strengthen the Conclusion:

Emphasize the public health implications of undetected co-infections and recombinant variants, potentially suggesting how surveillance systems could be improved.

In this manuscript, the authors create a large sequencing dataset from COVID-19 positive nasopharyngeal swabs with the aim of identifying SARS-CoV-2 viral populations in a patient, as a marker of co-infections and recombination events. While the sequencing data collected has significant potential and would be of interest to the field and the intended findings of this manuscript could be importance to the field, the manuscript would require significant changes to support some of the claims stated.

- 1) The manuscript claims that the method is a 'promising approach for the detection of co-infection events'. While this may be true, it relies solely on data collected in an uncontrolled manner. To more emphatically demonstrate the validity of the method the authors need perform co-infections in cell culture, sequence resulting RNA and repeat the methods as described.
- 2) Continuing from the above, the detection of two SARS-CoV-2 species from the same host is not equivalent to detecting two species in the same cell. Though both avenues have value, only the latter provides a direct insight into possible recombination.
- 3) For Figures 2 and 4 and the associated text in the body of the manuscript, the results are split into a timescale of months. The sectioning of the data in this manner does not serve a biological purpose and results in a loss of resolution. The results would instead be better represented on the scale of weeks which would also align with the study's weekly data collection and negate any potential issues with differing rounds of data collection (i.e. if data was collected on a Friday, not every month has the same number of Fridays).
- 4) For the VOC detection, the authors have provided the aa changes being searched for but not the corresponding nucleotide change(s). Additionally, given the sequencing method chosen generates short length reads, there is a possibility that detected viruses possess some but not all the mutations associated with VOC. This does not appear to have been accounted for.
- 5) There are several issues with the language used in the discussion including;
 - a. Some of the discussion reads more as a literature review/introduction and does not illustrate the findings of this study.
 - b. "to separate the samples based on their global diversity" - the samples were not globally diverse if they were all from France.
 - c. "This approach allowed us to generate key information for a better understanding of the transmission dynamics" – there is no associated data regarding transmission in this manuscript

Minor comments:

- 1) M&M – Study Design, final sentence. It is unclear as to whether the patient has been confirmed positive for COVID-19 by RT-PCR on the nasopharyngeal swab itself or by testing another sample from the patient.
- 2) M&M – Study Design, final sentence. Missing 'result' after RT-PCR.
- 3) Figure 3 – what is the threshold 29.05%

Response to Reviewers

Spectrum02092-24 - "Dual SARS-CoV-2 infections: Genomics-based approach for detection and characterization of SARS-CoV-2 co-infections and diverse viral populations"

Dear Editor: Thanks to the review process, we have revised the manuscript to address all of the reviewers' comments. We believe that the comments have greatly improved the manuscript and led to the addition of a new Results section. We hope that these changes have made the article more worthy of acceptance and we thank you for considering it for publication.

Reviewer 1

1) The manuscript claims that the method is a 'promising approach for the detection of co-infection events'. While this may be true, it relies solely on data collected in an uncontrolled manner. To more emphatically demonstrate the validity of the method the authors need perform co-infections in cell culture, sequence resulting RNA and repeat the methods as described.

Response: We agree with the reviewer that the article would benefit from a validation of the method. The good ethical and biosafety practices of our organization do not allow us to perform co-infections of viral agents such as SARS-CoV-2 in cell culture. Therefore, we generated an *in silico* cohort of co-infected samples at different ratios to test the method, as described in the Methods (line 405) and Results (line 145) sections of the revised manuscript. The synthetic data allowed us to identify thresholds and provide a stronger basis for the method to detect coinfection.

The threshold identified from the synthetic data to discriminate pure/very low co-infections from highly co-infected samples was found to be 29.0% (new Figure 3), identical to the threshold obtained in the original version of the manuscript by using samples with iSNV ratios significantly different from the background at $p < 0.01$. The results and figures are now presented in relation to the synthetic data results. We believe that this addition strengthens our methodology and makes the paper more coherent.

2) Continuing from the above, the detection of two SARS-CoV-2 species from the same host is not equivalent to detecting two species in the same cell. Though both avenues have value, only the latter provides a direct insight into possible recombination.

Response: The reviewer is correct. However, it is impossible to study coinfection/recombination at the cellular level in infected individuals. This limitation and the fact that coinfection favors, but does not automatically imply, recombination are now discussed in a dedicated paragraph in the Discussion (line 324).

3) For Figures 2 and 4 and the associated text in the body of the manuscript, the results are split into a timescale of months. The sectioning of the data in this manner does not serve a biological purpose and results in a loss of resolution. The results would instead be better represented on the scale of weeks which would also align with the study's weekly data

collection and negate any potential issues with differing rounds of data collection (i.e. if data was collected on a Friday, not every month has the same number of Fridays).

Response: We have changed the scale from months to weeks in both the figures and the text. We agree that the resolution of the results has improved, allowing a clearer view of some weekly results that were “masked” by the month format. In the Results section, the month format has been retained in brackets to give the results a time reference that can be compared with other reports.

4) For the VOC detection, the authors have provided the aa changes being searched for but not the corresponding nucleotide change(s). Additionally, given the sequencing method chosen generates short length reads, there is a possibility that detected viruses possess some but not all the mutations associated with VOC. This does not appear to have been accounted for.

Response: a) The appropriate nucleotide changes have been added to the Methods section (line 447). b) A more precise version of the Methods has been added to acknowledge the case where viruses have some but not all VOC mutations (line 455) and this has also been discussed in the “Limitations” section of the discussion (line 324).

5) There are several issues with the language used in the discussion including; a. Some of the discussion reads more as a literature review/introduction and does not illustrate the findings of this study. b. “to separate the samples based on their global diversity” - the samples were not globally diverse if they were all from France. c. “This approach allowed us to generate key information for a better understanding of the transmission dynamics” – there is no associated data regarding transmission in this manuscript

Response: a) The discussion has been revised and the information from the literature not directly related to the results of the study has been removed or adapted to better fit the focus of the study. In addition, the discussion has been expanded to better reflect the findings and to avoid a “literature review” effect. b) The word “global” has been changed to “genomic” to avoid misunderstanding. c) The sentence has been corrected by removing “transmission dynamics.”

In addition, the revision of the article allowed us to improve the writing quality in several sentences, mainly in the Introduction and Discussion sections. All these changes have been annotated in the marked-up version.

Minor comments:

1) M&M – Study Design, final sentence. It is unclear as to whether the patient has been confirmed positive for COVID-19 by RT-PCR on the nasopharyngeal swab itself or by testing another sample from the patient.

2) M&M – Study Design, final sentence. Missing ‘result’ after RT-PCR.

3) Figure 3 – what is the threshold 29.05%

Response: All changes suggested in the “minor comments” have been implemented and the explanation of the 29.0% threshold has been added in the Methods and in Figure 3 by adding the validation using synthetic data.

Reviewer 2

1. Clarity and Detailed Description of Methodology:

iSNV Detection and Threshold Determination: The methods section would benefit from a more comprehensive explanation of how iSNVs were identified and the rationale behind choosing the specific thresholds (e.g., frequency greater than 5%, $p < 0.01$ for iSNV ratios). Providing details on how the background noise was calculated and how the threshold of 29.05% was established would enhance reproducibility and understanding.

Response: This was a very important point for the revision of the manuscript, also discussed in Reviewer #1's comments. To better address this and to validate the method, we generated a synthetic cohort that we included in the threshold determination. The inclusion of the synthetic cohort to validate the iSNV method and to obtain more accurate thresholds based on iSNV ratios of true co-infections provided a stronger basis for how we identified potentially co-infected samples. The rationale has been explained in detail in the new version of the manuscript, in Methods (line 405) and Results (line 145).

Confirmation of Co-infections: The criteria for confirming co-infections based on VOC-specific mutations in the spike protein should be elaborated. Discussing the potential for false positives or negatives when only a subset of mutations is used for confirmation would strengthen the validity of the findings.

Response: Regarding the confirmation of co-infections, we have elaborated on the criteria for the use of the spike protein (line 442) and we included more details on the subset of mutations used and the criteria for confirmation (line 455). We have also addressed the potential for false positives and negatives in the new "Limitations" section (line 324).

2. Statistical Analysis and Interpretation:

Significance Testing: More detailed information on the statistical methods used to assess the significance of observed iSNV ratios would be valuable. For instance, explaining the statistical tests applied and any assumptions made would improve the robustness of the results.

Response: In the new Methods and Results, more details have been added to explain the threshold determination (iSNV ratio limits for 99% of highly co-infected samples and 99% of pure and low co-infected samples) and a ROC curve has been plotted to validate the sensitivity of the method (Methods line 405, Results line 150 and Figure 3).

Co-infection Rates Calculation: Clarify how the co-infection rates were calculated concerning the total number of cases during co-circulation periods. Including confidence intervals would provide a sense of the precision of these estimates.

Response: Details on how the co-infection rates were calculated have been added to the Methods (line 456). As we report the co-infection rates of our cohort based on the co-infections observed in relation to the total number of cases in the cohort, we did not calculate any estimates.

3. Discussion of Limitations:

The manuscript would benefit from a more in-depth discussion of the limitations inherent in the study. For example, acknowledging potential biases due to sample selection, limitations in detecting co-infections with closely related lineages, and the impact of sequencing depth and quality on iSNV detection would provide a balanced perspective.

Response: We agree with the reviewer. A specific paragraph discussing the limitations of our study has been added to the Discussion (line 324).

Consider discussing the potential influence of immunocompromised patients on iSNV rates, as they may exhibit higher intra-host diversity, potentially confounding the detection of co-infections.

Response: A more detailed discussion of immunocompromised patients and intra-host diversity has been added to the Discussion (line 258).

4. Data Accessibility and Transparency:

While the authors mention that sequences were submitted to the GISAID database and that raw data are available upon request, providing accession numbers or a direct link to the dataset would enhance transparency and allow other researchers to verify and build upon the work.

Including supplemental materials with additional data tables, such as a detailed list of confirmed co-infection cases and their associated mutations, would be beneficial.

Response: The accession numbers have been added to the Methods section (line 386). A supplemental table with the list of mutations of the confirmed co-infections has been included.

5. Contextualization with Existing Literature:

The discussion could be enriched by comparing the co-infection rates found in this study with those reported in other regions or countries, exploring reasons for similarities or differences.

Incorporating recent literature on recombinant variants and their impact on public health would emphasize the relevance of the findings.

Response: The Discussion has been modified to include more recent literature on recombinant variants, and to better highlight the findings in comparison to other co-infection rates reported in studies from other countries. In addition, based on a comment from Reviewer #1, we have also modified the discussion to read less like a literature review and more like an evaluation of our findings.

Minor Comments:

1. Consistency in Terminology:

Ensure consistent use of terms such as "co-infection," "dual infection," "mixed infection," etc., throughout the manuscript to avoid confusion.

Response: Throughout the manuscript, all alternative terms have been changed for "co-infection" to ensure consistency.

2. Enhance the Introduction:

Provide a more detailed background on the mechanisms of SARS-CoV-2 recombination and why co-infections are pivotal in this context. This would set a stronger foundation for the study's significance.

Response: An additional paragraph on the mechanism of coronavirus recombination has been added to the Introduction (line 74).

3. Strengthen the Conclusion:

Emphasize the public health implications of undetected co-infections and recombinant variants, potentially suggesting how surveillance systems could be improved.

Response: The conclusion has been expanded to include an emphasis on public health implications, surveillance systems, and the overall relevance of the study (line 344).

Re: Spectrum02092-24R1 (Genomics-based approach for detection and characterization of SARS-CoV-2 co-infections and diverse viral populations)

Dear Dr. Bryan Jimenez-Araya:

Your manuscript has been accepted, and I am forwarding it to the ASM production staff for publication. Your paper will first be checked to make sure all elements meet the technical requirements. ASM staff will contact you if anything needs to be revised before copyediting and production can begin. Otherwise, you will be notified when your proofs are ready to be viewed.

Sincerely,
Day-Yu Chao
Editor
Microbiology Spectrum

Reviewer #2 (Comments for the Author):

The authors have addressed all my concerns. I have no more questions.